# Superabsorbent Polymers as a Soil Amendment for Increasing Agriculture Production with Reducing Water Losses under Water Stress Condition

**DOI:** 10.3390/polym15010161

**Published:** 2022-12-29

**Authors:** Shweta Malik, Kautilya Chaudhary, Anurag Malik, Himani Punia, Meena Sewhag, Neelam Berkesia, Mehak Nagora, Sonika Kalia, Kamla Malik, Deepak Kumar, Pardeep Kumar, Ekta Kamboj, Vishal Ahlawat, Abhishek Kumar, Kavita Boora

**Affiliations:** 1Department of Agronomy, College of Agriculture, CCS Haryana Agricultural University, Hisar 125004, Haryana, India; 2Department of Seed Science & Technology, College of Agriculture, CCS Haryana Agricultural University, Hisar 125004, Haryana, India; 3Chandigarh Group of Business, Department of Agriculture, Chandigarh Group of Colleges, Jhanjeri, Mohali 140307, Punjab, India; 4Department of Biochemistry, College of Basic Sciences & Humanities, CCS Haryana Agricultural University, Hisar 125004, Haryana, India; 5Chandigarh Group of Business, Department of Sciences, Chandigarh Group of Colleges, Jhanjeri, Mohali 140307, Punjab, India; 6Department of Microbiology, College of Basic Sciences & Humanities, CCS Haryana Agricultural University, Hisar 125004, Haryana, India; 7Department Soil Science, College of Agriculture, CCS Haryana Agricultural University, Hisar 125004, Haryana, India; 8Department Pathology, College of Agriculture, CCS Haryana Agricultural University, Hisar 125004, Haryana, India

**Keywords:** super absorbent polymers, agriculture, hydrogel, water scarcity, polymerization

## Abstract

With an increasing population, world agriculture is facing many challenges, such as climate change, urbanization, the use of natural resources in a sustainable manner, runoff losses, and the accumulation of pesticides and fertilizers. The global water shortage is a crisis for agriculture, because drought is one of the natural disasters that affect the farmers as well as their country’s social, economic, and environmental status. The application of soil amendments is a strategy to mitigate the adverse impact of drought stress. The development of agronomic strategies enabling the reduction in drought stress in cultivated crops is, therefore, a crucial priority. Superabsorbent polymers (SAPs) can be used as an amendment for soil health improvement, ultimately improving water holding capacity and plant available water. These are eco-friendly and non-toxic materials, which have incredible water absorption ability and water holding capacity in the soil because of their unique biochemical and structural properties. Polymers can retain water more than their weight in water and achieve approximately 95% water release. SAP improve the soil like porosity (0.26–6.91%), water holding capacity (5.68–17.90%), and reduce nitrogen leaching losses from soil by up to 45%. This review focuses on the economic assessment of the adoption of superabsorbent polymers and brings out the discrepancies associated with the influence of SAPs application in the context of different textured soil, presence of drought, and their adoption by farmers.

## 1. Introduction

In agriculture, water utilization is 85% compared to industrial (15%) and domestic (5%) use. Thus, emphasis should be given to water conservation, with scarcity affecting crop productivity. There is a need for efficient water resources for reducing water losses [1,2]. Drought is among the worst natural disasters that affect the whole world. Drought leads to water distress and causes poor irrigation practices and water management, dreadful conditions of soil quality, along with low water holding capacity. Water sapping or water crises have been reducing yields in numerous field crops for a long time. Mazloom et al. [3] reported a drop of up to 21% in wheat yield and up to 40% in maize yield owing to worldwide water scarcity. According to the IPCC’s 2014 report, climate change is increasing drought stress as temperatures rise. Water stress makes crops more vulnerable than those that grow in arid or semi-arid environments. This feature has caused concern in recent decades, as exceptional and harsh weather events have hampered crop yield in tropical and subtropical climate zones. As a result, a drought scenario can be defined as a circumstance in which plants are unable to take water from the soil and are subjected to water stress [4,5]. 

Due to climatic changes, water resources are more affected [6,7,8,9]. Developing countries still rely on rainfed agriculture. Any effort to improve water use efficiency in agriculture is worthwhile. In India, the dryland area covers approximately 60% of the net cultivated. India has ranked 41 among 181 countries in the World [10] for water crises. In semi-arid conditions, the main causes of low agriculture productivity are water losses by more evaporation and transpiration, leaching of soil water, and less retention capacity and water stress cause less growth and yield [11,12,13,14,15,16,17]. Under high temperatures, the risk of drought increases, low photosynthesis, and rapid development of the plant might be due to less CO_2_ assimilation and light interception [18,19,20].

Agricultural technologies have been created to increase agricultural output while simultaneously lowering resource costs and environmental losses associated with agriculture. Rainwater harvesting, reservoirs, and streamflow diversion are used to meet the demand for and supply of water [21]. By using these technologies, we can conserve natural resources by the application of fewer inputs while maintaining an optimum level of yields. Other ways to conserve water involve improving the water-holding capacity of the soil, minimizing water percolation losses, and reducing irrigation requirements. Soil improvement is the best way to improve water and nutrient retention in the soil, as well as reduce infiltration and drainage losses, resulting in improved plant development [22]. Super-absorbents or hydrogel may be helpful to increase the quantity of moisture available in the root zone of crops because polymers can absorb water 400–1500 times their dry weight [23,24,25,26,27]. SAP absorb large quantities of water and solute because of various hydrophilic groups attached to their polymeric backbone viz. carboxyl, amino, and hydroxyl groups [28]. Polymers do not reduce water consumption but, due to the cross-linked polymers, can retain water more times its weight and achieve approximately 95% water release [29,30,31]. Under water stress conditions, plants could grow properly with the hydrogel [32]. Nanocomposite hydrogels have been utilized to improve water and nutrient retention in agricultural soils as an alternative to the existing needs of the agricultural sector [33]. Nanoparticles and nano-layers have a high surface-to-volume ratio, making them ideal for usage in a network of polymeric materials [34]. This review summarizes the diverse applications of polymers in intensive agricultural systems that minimize fertilizer losses along with lower water requirements and minimal effects on quality and yield. The interpretation of agricultural characteristics (i.e., boost NUE and resilience towards abiotic stresses, purifiers of water) would allow the designing of preparation of second-generation polymers in which synergistic and compatible processes may be practically developed and implemented in future studies.

## 2. Water Stress Status and Irrigation

Around 2.3 billion people live in water-stressed countries, of which 733 million live in high and critically water-stressed countries. Moreover, 3.2 billion people live in agricultural areas with high to very high water shortages or scarcity, of whom 1.2 billion people, roughly one-sixth of the world’s population, live in severely water-constrained agricultural areas [35]. The impacts of a changing climate are making water more unpredictable. Terrestrial water storage, i.e., the water held in soil, snow, and ice, is diminishing. This results in increased water scarcity, which disrupts societal activity. Soil erosion from croplands carries away 25–40 billion tonnes of topsoil every year, significantly reducing crop yields and the soil’s ability to regulate water, carbon, and nutrients, and transporting 23–42 million tonnes of nitrogen and 15–26 million tonnes of phosphorus off the land, with major negative effects on water quality. Naturally occurring arsenic pollution in groundwater now affects nearly 140 million people in 70 countries on all continents. 

India has scored 4.2 with regard to water stress on the 0–5 scale system and is the 41st rank among all countries [36]. This indicates that India comes under the high-risk zone [36]. In the future, India is also going to face a significant water crisis. In underwater scarcity conditions, the per capita availability becomes below 1000 m^3^ (Table 1). Therefore, there is a requirement for efficient water management by ameliorating water use efficiency. The important sector is agriculture for saving water without affecting crop yield [37]. Table 1 predicted that during 2011 demand was less than the per capita availability of water. Up to 2050, per capita, water availability would be less and water demand would be more, with maximum water required for irrigation (Report, NCIWRD, 1999).

‘To boost up crop production and performance applying the water in crops is known as irrigation’ [38,39]. Under rainfed areas, landscape, sowing, germination of seeds, and life-saving water supply depend on rainfall [40]. The regular practice of irrigation has moved from basin and furrow to micro-irrigation types. Nowadays, priority has to be given to yield per drop of water rather than per unit area as considered earlier [41,42]. Uncontrolled irrigation is one of the major issues for lack of water management [43,44]. Hydrogels are the most efficient way to improve crop irrigation as compared to traditional irrigation (Table 2).

## 3. How Is Superabsorbent Polymer Created?

Water or other aqueous fluids can be absorbed by superabsorbent polymers, which are macromolecular cross-linked hydrophilic polymer chains [45]. Super absorbent polymers (SAPs) are materials with the enormous liquid holding capacity compared to their size. Superabsorbent polymers are natural or manufactured polymers that can absorb a high percentage of water (Figure 1). Polymeric materials that display the capacity to expand in water and retain a considerable portion (>20%) of water inside their structure without dissolving in water are known as hydrogels. They also possess a degree of flexibility very similar to natural tissue due to their large water content. When the polymer is placed in the soil, it absorbs water from irrigation and rainfall, functioning as an extra water reservoir.

The absorbed water is then released as the soil dries, making it available to the plants [4]. The swelling of the polymer in the soil improves porosity, air capacity, and CEC while decreasing the infiltration rate [47]. 

### Hydrogel Preparation Technologies Used

Hydrogels are polymer networks with hydrophilic characteristics. Hydrophilic and hydrophobic monomers are sometimes utilized in the manufacture of hydrogels to manage the characteristics for certain purposes. Hydrogels can be created by copolymerizing/cross-linking free-radical polymerizations of hydrophilic monomers and multifunctional cross-linkers. Water-soluble linear polymers of both natural and synthetic origin are crosslinked in a variety of ways to generate hydrogels.
Ionizing radiation is used to create main-chain free radicals that can recombine as cross-link junctions.Using a chemical process to connect polymer chainsEntanglements, electrostatics, and crystallite formation are examples of physical interactions. Monomer, initiator, and cross-linker are the three essential components of hydrogel production.
Bulk polymerization: For the creation of hydrogels, Bulk hydrogels can be generated with one or more types of monomers, the most common of which being vinyl monomers. In most hydrogel formulations, a tiny quantity of crosslinking agent is included. The polymerization reaction is initiated by radiation, ultraviolet light, or chemical catalysts. The initiator is chosen based on the kind of monomers and solvents employed. Polymerized hydrogels may be manufactured in a number of shapes, including rods, particles, films and membranes, and emulsions.Free radical polymerization: The primary monomers employed in this approach for the manufacture of hydrogels are acrylates, vinyl lactams, and amides. These polymers include functional groups that are appropriate for polymerization or have been functionalized with radically polymerizable groups. The chemistry of conventional free radical polymerizations is used in this approach [8,9,14], which comprises propagation, chain transfer, initiation, and termination phases. A wide range of thermal, ultraviolet, visible, and redox initiators can be used to generate radicals in the initiation stage; the radicals react with the monomers, converting them into active forms.Solution polymerization/cross-linking: The multifunctional crosslinking agent is combined with these ionic or neutral monomers (Figure 2). UV-irradiation or a redox initiator method initiates the polymerization thermally. To remove the initiator, soluble monomers, oligomers, cross-linking agents, extractable polymer, and other contaminants, the hydrogels are washed with distilled water. Water-ethanol combinations, water, ethanol, and benzyl alcohol were utilized as solvents.Suspension polymerization or inverse-suspension polymerization: The advantage of this method is that the products are obtained as powder or microspheres (beads). Thus, grinding is not required. The monomers and initiators are disseminated as a homogeneous mixture in the hydrocarbon phase using this approach. The size and form of the resin particles influence the viscosity of the monomer solution, rotor design, agitation speed, and dispersant type. The dispersion is thermodynamically unstable and needs both constant agitation and the addition of a suspending agent with a low hydrophilic-lipophilic balance (HLB).Grafting to support: Because of the fragile structure of hydrogels created by bulk polymerization, it is important to increase a hydrogel’s mechanical qualities so that it may be surface coated onto stronger support. This entails generating free radicals on a stronger support surface and then directly polymerizing monomers onto it to generate a chain of monomers that are covalently bound to the support.Polymerization by irradiation: In the creation of unsaturated compound hydrogels, initiators such as ionizing high energy radiation, such as gamma rays and electron beams, have been utilized. Irradiating an aqueous polymer solution causes radicals to develop on the polymer chains. Irradiation polymerization uses poly (vinyl alcohol), poly (ethylene glycol), and poly (acrylic acid). This approach yields hydrogels that are quite pure and devoid of initiators.Physical cross-linking: It is made by chilling heated gelatin or carrageenan solutions to generate physically cross-linked gels. The gel is formed as a result of helical association, helix creation, and the production of junction zones. Polyethylene glycol-polylactic acid hydrogel and polyethylene oxide-polypropylene oxide are two examples. It is the most popular and straightforward method for forming hydrogels by cross-linking polymers via physical interactions. Ion interactions, such as hydrogen bonding, polyelectrolyte complexation, and hydrophobic association, are examples of physical cross-linking. The following procedures are used to create physically cross-linked hydrogels:
Heating/cooling a polymer solution: It is made by chilling heated gelatin or carrageenan solutions to generate physically cross-linked gels. The gel is formed as a result of helical association, helix creation, and the production of junction zones. Polyethylene glycol-polylactic acid hydrogel and polyethylene oxide-polypropylene oxide are two examples.Complex coacervation: Polyanions and polycations are mixed to form complicated coacervate gels. This method’s core idea is that polymers with opposing charges cling together and create soluble and insoluble complexes depending on the concentration and pH of the corresponding solutions. Coacervating polyanionic xanthan with polycationic chitosan is one such example.Ionic interaction: Cross-linking between polymers occurs when divalent or trivalent counter ions are added to an ionic polymer. This approach is based on the gelling polyelectrolyte solution concept.

## 4. Nano-Irrigation for Solving the Irrigation Problem in Agriculture

In agriculture, nanotechnology is measured as very important [48] for crop and apparent water productivity [49]. In the world, various technology, devices, and systems beneficial to farmers are available. They are used in irrigation, water purification, carbon nanotubes nano-enabled membrane filters, and ceramic and magnetic particles for water management. Irrigation water can be treated with some available devices, e.g., nano-8630 for improving crop performance [6,8,9,14,16,50,51,52,53,54]. Different methods, such as bench terracing, counterbidding, and micro-irrigation systems as ex-situ methods and tillage (Zero, Minimum, and Conservation), cultural practices (furrow and ridge sowing), and some chemicals, such as anti-transpirants and super absorbent polymers, can be used as in-situ for preserving and shrinking water application in the field. The uses of polymers are important where water scarcity occurs [55,56]. Polymers are non-toxic and decompose easily without any residual impact. Polymeric soil conditioners help to keep aggregates stable by gluing particles together within them and coating the particle surfaces [57]. Synthetic soil conditioners have a lot of potential for enhancing soil productivity where irrigation isn’t an option. It was revealed that in sandy soil treated with gel conditioner, water storage at various tensions improved dramatically. Satriani et al. [58] reported that by utilizing SAP hydrogel for the purpose of conserving irrigation water, it is possible to keep the production level the same while simultaneously decreasing the quantity of irrigation water used. A healthy soil water potential was preserved by the use of a super absorbent hydrogel that was applied to the plants of a local bean population that was experiencing the adverse effects of drought.

## 5. Super Absorbent Polymers Application in the Soil for Agriculture

Irrigation processes are costly and require maintenance of optimum moisture levels in soil with enough supply of nutrients and water to plants. Hence, certain polymers have become popular to avoid these problems and to reduce the cost of incessant irrigation procedures. Some of the polymers are water-soluble and some are not soluble. It was discovered that adding polymer to sandy soils increased plant water availability by increasing retention pores and decreased saturated hydraulic conductivity by reducing drainage pores [59]. Hydrogel application in sandy soils greatly boosted water retention capacity, according to other studies [29,60], while the effect of hydrogel was negligible in loam and clay soils. However, several investigations have come up with contradictory conclusions. According to Leciejewski [61], soil water storage increased mostly in the pFb2 range in sandy soils treated with hydrogel. These results were similar to those reported earlier by Paluszek and Zembrowski [59]. In the gel-treated plot, essential SWC arrived early (4–7 days) and water transmission pores/aeration pores were severely reduced (much below the critical limit of 10% aeration porosity). To grow vegetables in alluvial and red sandy loam soils, hydrogels were found to enhance water availability to plants by 1.5–2 times over the water accessible to plants grown in non-gel-treated soils. Moreover, 7–15 days of irrigation corresponded to the onset of key SWC (7–14 DAW) in the field. It was determined that hydrogel was an excellent medium for the cultivation of agricultural crops on sandy soils because the amount of time it took for the soil to reach its critical SWC after being treated with hydrogel was almost 22 days, which is the same amount of time between irrigations that is required for the majority of agricultural crops [62]. 

## 6. Water-Soluble Polymers and Their Advantages

Water-soluble polymers are used to aggregate soil, and reduce percolation losses, and soil erosion. Some are polyacrylates, vinyl alcohol, polyacrylamide, and poly-vinyl acetate-alt-maleic polymers used as soil conditioners. Mostly polymers are synthesized through free radical polymerization, except ethylene glycol. Polymers should have high molar mass.

They are effective against soil erosion and reduce soil loss under heavy rainfall situations. Water penetration increases in polymer-treated soil according to several studies. Because of soil aeration, microbial activity is enhanced, and the uptake of nutrients by plants is increased [63,64].

## 7. Gel Forming Polymers

### 7.1. Pusa Hydrogel

Pusa hydrogel is a natural polymer based on cross-linked potassium polyacrylate polymer in soil. It is stable for a minimum period of one year, less affected by salts, and has no adverse effect on crops. A semi-synthetic high-capacity super absorbent polymer based on starch from CTCRI used cassava starch backbone.

### 7.2. Alsta Hydrogel

Alsta hydrogel is a potassium polyacrylate-based polymer. It can absorb 300–500 times water and release it accordingly. It helps reduce irrigation frequency. It can be used in hydroseeding, hydroponics open fields, and protective cultivation [65]. 

### 7.3. Characteristics of Hydrogel

The high absorption capacity of the hydrogel is achieved with the lowest soluble content and residual monomer, low price, high durability and stability, biodegradability without any toxin formation, photostability with the rewetting capability, and loose, granular, and powdery appearance. The shelf life of gel is 2–5 years in soil [10].

### 7.4. Advantages of Agriculture Hydrogel

Agriculture hydrogel improves soil quality, preserves water, and resists drought, offers better seed sprouting and seedling development, reduces irrigation frequency and water consumption, and creates a cyclic process to provide water directly to roots and prevents soil compaction. It acts like a micro water reservoir at plant roots and can absorb water 400–500 times its weight and release it slowly on account of the root capillary suction mechanism, thus preventing water loss in soil by leaching and evaporation. It provides optimum moisture for quick seedling germination, maturation, and root growth and density [66,67,68,69]. 

The application of polymers enhances the O_2_ accessibility in the plant root zone, due to microbial activity enhancement [70,71]. This helps the crops to resist the permanent wilting point and continue to exist under excess moisture stress. Polymers minimize evaporation losses and irrigation water requirements of plants. The nutrient losses occur through leaching with less water and water held in the crop root zone [72,73,74].

### 7.5. Guar Gum Polymers (Organic Hydrogel)

Guar gum is an edible carbohydrate, known as galactomannans. It is also capable of producing more gummy, pseudoplastic, aqueous conditions at very low concentrations. Guar gum can be modified by oxidation, enzymatic hydrolysis, etherification, cross-linking, esterification, and graft depending on its applicability [24].

### 7.6. Characteristics of Guar Gum Polymers

Guar gum is a stabilizing agent, and emulsifier, having film-forming properties, flocculation, and maintaining high viscosity and pH. Guar gum increases the water-holding capacity and compactness of the soil. It reduces soil erosion and runoff losses. It also increases the infiltration and permeability of soil [24,75].

Like an agrochemical, it does not affect nutrient availability and soil chemical composition due to neutral pH. It improves the soil’s physical properties, such as porosity, bulk density, water holding capacity (WHC), soil permeability, and infiltration rate [60,76,77]. It also maintains soil fertility, taking up an incredible amount of water. The commercial availability of hydrogel with trade names and manufacturing group names available in India is shown in Table 3.

### 7.7. Hydrogel-Biochar

In this respect, using nano-biochar/natural char as a substructure material for the creation of composites has a number of benefits over using other composite materials, such as greater performance and lighter weight [72,73,78]. NCNPs were made via a chemical process. Briefly, potassium permanganate, a potent oxidizer, oxidized finely-ground NC powder in an acidic environment. The oxidation reaction’s precipitate was then washed with deionized water, centrifuged, and dried.

### 7.8. Agronomic Applications

Polymers are applied to build up the soil’s capacity to absorb water. Polymers are primed by grafting and crosslinking of polyacrylamide (water-absorbent polymers) onto carboxymethyl cellulose (cellulose derivative backbone polymer chain). They are biodegradable and environmentally friendly [60]. The hydrogels applied in the agriculture field should not only have the ability to take in water but must also liberate the same slowly according to the specific necessity of the plants [74,75,79,80]. The application of hydrogels can result in a significant reduction in the required irrigation frequency in coarse-textured soils and it is important in coarse-textured soils of arid and semi-arid areas of the world for enhancing water management [81]. In Eygpt, it was found that in wheat crops hydrogel saves more moisture in the root elongation zone [82]. Hydrogel maintains a continuous moisture supply at the root zone for good root establishment and various plant processes [83,84,85,86]. In fodder sorghum, 5 kg/ha of hydrogel application statistically increases MC (moisture content) of soil at different depths, viz. 0–15, 15–30, and 30–45 cm, at different crop growth stages [87,88]. In sandy soils, hydrogel increased the different enzymatic activities, e.g., dehydrogenase, acid phosphatase, urease, alkaline phosphatase, and protease in the soil [24,89].

Agricultural hydrogel application rates vary from soil to soil and are −2.5 kg/ha for clay soil up to the depth of 6–8 inches and up to 5.0 kg/ha in sandy soil (at the soil depth of 4 inches) [10]. In India, different wheat experiments under diverse wheat zones (viz. northeastern plain zone, central and peninsular zone) illustrate that under different irrigation levels (viz. no irrigation, two times irrigation, and four times irrigations) hydrogel with the rate of 5 kg/ha application created the statistically maximum yield. The use of four-time irrigation without hydrogel gave equivalent to two sessions of irrigation along with 5 kg/ha hydrogel wheat yield [90]. The soybean yield was increased by the application of hydrogel [91], and a higher leaf area was found in the case of Sweet Pepper with the application of hydrogel [92]. 

Aerobic rice produced significantly superior growth and yield attributes and rice yield with hydrogel application of 2.5 kg/ha compared to control in all kinds of sowing methods, viz. flatbed sowing, ridge sowing, and raised bed sowing [81]. 

Seed treatments with hydrogel at the rate of 10 and 20 g/kg seed illustrate the significant maximum yield attributes viz., tillers/m^2^, ear length (cm), 1000- grain weight, and economic yield compared to control and water soaking [93]. Under clay loam soil, the wheat yield was reported 8.48% more in 5 kg/ha hydrogel application over to control with 100% fertilizer dose [89]. The growth parameter of the crop increased under available adequate moisture [94,95]. In Peanut, hydrogel at a rate of 200 kg/ha was reported as statistically better for yield and attributes, viz., economic yield, biomass, number of pods per plant, branches per plant, and 100- seed weight at sandy soil in the hot and arid climate of Iran [81]. Wheat varieties, at different locations of Uttar Pradesh, under hydrogel with the rate of 5 kg/ha along with three times irrigation are capable to generate an equivalent grain yield to five times irrigation without hydrogel. The soil treatment with hydrogel can accumulate two irrigations without reducing the wheat yield [96]. The grain yield of wheat and soybean was increased due to hydrogel application [97,98,99]. 

It was reported that hydrogel granule size has a negative interaction with grain yield, as granule size increases and wheat grain yield reduces [100]. The application of hydrogel at different rates also affects the productive and non-productive tillers of crops and similar treads are also observed in chrysanthemums [1]. Irrigated wheat crops had low nitrogen content in grain [101]. Nitrogen and potash content in wheat grain can be increased by reducing the irrigation frequency [94,95,102]. P content was observed more under higher irrigation comparison to no irrigation [103,104]. In wheat (Sandy clay loam soil) crops under limited water, two irrigations with hydrogel with 7.5 kg/ha produced high yield attributes along with nutrient uptake [105,106]. In all types of soil along with all crops, polymers can be applied. It is mainly beneficial for nurseries, seedlings, and moisture-sensitive crops, as they require a high amount of water and a gardener or pot grower. From the study, it was observed that hydrogel has a significant effect on pearl millet yield and harvest index (Table 4) [1]. 

### 7.9. Role of the Superabsorbent in Soil Properties

Soil physical index (S) was higher with hydrogel means more pores due to the formation of new aggregates [110]. A value above 0.035 of “S” means soils have good physical quality [111,112,113]. Due to the natural pH of the hydrogel, they did not affect the nutrient availability to plants, and there was no exploitation of other chemicals. Table 4 predicts that the application of hydrogel improves soil properties. Hydrogel has the potential to improve the soil like porosity, and water holding capacity [68,114], improve seed germination, and root growth, reduce soil erosion losses, as well as improve microbial activity so a supply of O_2_ improves the root zone [27]. The study conducted in Delhi, India indicates that hydrogel application reduces the number of irrigations in wheat crops compared to without hydrogel application (Table 5) and indicates that under moisture stress hydrogel plays an important role in enhancing grain yield, improving water use efficiency [62,71,100,115]. A significant benefit of soil hydrogels is their gradual release, which allows for the vital nutrients to be released from the hydrogel matrix at a rate that allows a plant to make use of them for a more extended period of time [116]. The plants that were cultivated in soil that had been treated with hydrogel exhibited enhanced physiological and morphological features, as well as higher survival, water-use efficiency, and dry matter production [117,118]. Satriani et al. [58] conducted a study to find out the role of superabsorbent hydrogel in bean crop cultivation under deficit irrigation conditions, and apply hydrogel with drip irrigation and observed that the agricultural water productivity index can be maximized by combining hydrogel soil amendments with deficit irrigation. In point of fact, the soil amendment hydrogel techniques that were used in settings of water deficit irrigation yielded the highest water usage efficiency indexes. The findings may be helpful in maximizing the efficiency with which water resources are utilized in bean crop cultivations throughout the Mediterranean regions. The physical and chemical crosslinking of polymeric chains with various nano-scaled structures results in a network with novel properties, e.g., significant water retention and the ability to withstand large changes in pH, temperature, and ionic strength of the swelling solution [116,119]. Mikkelsen et al. [50] discovered that the addition of polymer to the fertilizer solutions reduced nitrogen leaching losses from soil columns by as much as 45% during the first four weeks in heavily leached conditions when compared with N fertilizer alone. At the same time, fescue (*Festuca arundinacea* L.) growth was also increased by as much as 40%, and tissue nitrogen accumulation increased by as much as 50% when fertilized with polymer rather than fertilizer alone. In addition, Magalhaes et al. [120] discovered that the presence of the polymer resulted in a significant reduction of the leaching of NH_4_, P, and K. Islam et al. [24] also show a noteworthy increase in soil nutrients in the presence of SAP, which may be attributable to a reduction in the leaching loss of those soil nutrients.

### 7.10. Soil-Plant Superabsorbent Interaction

Interactions between soil, plants, and superabsorbents can be affected by the properties of the plants, the soil, and the superabsorbent. The goal of this is to gain an understanding of the impact SAP has on a variety of soil variables (such as water holding capacity behavior, plant water supply content, hydraulic conductivity, and infiltration characteristic) as well as a variety of plant parameters (including root development, cell elongation, and yield components), as well as the relevance of these factors for agricultural production in drought management (Figure 3).

Previous research has shown that the addition of SAH to soil significantly improves its capacity to retain water, which is a desirable property in agricultural settings [124,125,126,127,128]. 

The SWC is the amount of water at which the soil is continuously saturated. The minimal water content at which there is no discernible change in water content with suction is indicated by the RWC, and the AEV indicates the suction value at which air enters the biggest pore of the soil matrix. Saturated water content (SWC), air entry value (AEV), and residual water content are crucial to WRCC. Incorporating the findings from this study and the literature showed that the PAWC improvement factor is principally texture specific and can be calculated purely from the WRCC of a specific soil, regardless of the type of plant. In times of water stress, it slows down the pace at which water evaporates from the soil [125,129,130,131]. The water inside the SAP is gradually released to the earth when the water outside the SAP has evaporated completely [71]. It is a well-known fact that soil salinity negatively affects plant water intake because the soil retains more water under osmotic pressure. By binding the salt ions in the long chain structure of the polymer, the addition of SAP in saline soil can reduce the pore water’s salt content. In addition, the hydrophilic polymers’ salt ion concentrations are reduced. According to Zhang et al. [132], the presence of salt cation reduces the absorption capacity of SAP by causing the gel network to produce more ionic crosslinks. The polyacrylate chain was shown to disintegrate in the soil at a relatively slow rate of 0.12–0.24% per six months, and the rate did not change much when the temperature rose. A biodegradable polymer with qualities that are comparable to those of commercial SAH is also the subject of contemporary research. These naturally occurring waste products are used to create these laboratories-produced SAHs. Some examples include starch [132,133,134], cellulose [71,135,136], chitosan [132,137] yeast [28], and clay powder [138,139]. 

### 7.11. Mode of Degradation of Bio-Polymer-Based Superabsorbent Polymers

Biopolymer-based membranes are resistant to hydrolytic degradation. The hydrogel degradation occurs after approximately six months [140,141]. Some studies observed that hydrogels are sensitive to UV rays and degrade into oligomers, and annually 10–15% can degrade into H_2_O, CO_2_, and N compounds. Biodegradable polymers degrade by microorganisms (enzymatic action of fungi, bacteria, and algae) and chemical hydrolysis processes [64,142,143]. A number of studies have shown that laboratory-made SAH performs better than commercial superabsorbent polymers in terms of water absorption capacity, salt sensitivity, and swelling ability [132,144,145,146,147,148,149]. A starch-modified poly (acrylic acid) SAH degraded at a rate of 40% after three months, according to Sarmah and Karak [150], who used the approach of soil burial.

## 8. Conclusions

In the case of almost all the crops, yield increases with the application of superabsorbent polymers. It also improves the product quality, biological environment, and hydro-physical properties of the soil. Its application remains dependent upon crop, environment, and region, Hence, superabsorbent polymers by seed treatment or soil application may increase agriculture productivity to enhance soil aeration in a water-stressed environment. SAP amendment was discovered to be a dependable remedy for guarding against long-term water stress conditions in the soil and plant system. The application of SAP decreased soil evaporative loss and water percolation into the deeper layer, which limited the amount of water available to the root zone. SAPs present a wide area of applications ranging from agriculture and forestry, industrial planting, municipal gardening, and drought management, to water conservation. It helps reduce soil erosion by surface run-off, fertilizer, and pesticide leaching to groundwater, thus reducing water, and irrigation costs while increasing the growth rate and high yield of crops. Economic assessments of the adoption of superabsorbent polymers are still lacking and may stimulate further adoption by farmers.

## 9. Future Perspectives

To date, it is not clarified how many times it takes to achieve degradation in the field. Moreover, it has also not been studied how many times it can absorb the water and if applied in a previous crop, whether it can affect the next crop or not. Responses under different tillage conditions have also not been studied. Thus, in the future, the relationship between tillage and soil hydrogel applications should be studied.

## Figures and Tables

**Figure 1 polymers-15-00161-f001:**
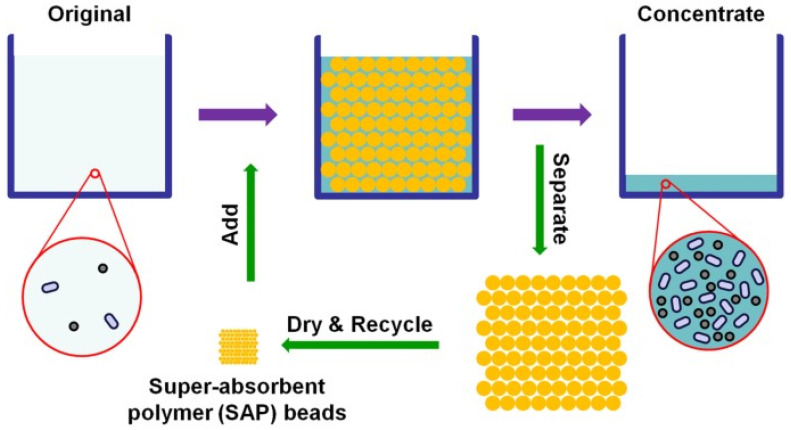
Schematic diagram of super-absorbent polymer (SAP) beads preparation [46].

**Figure 2 polymers-15-00161-f002:**
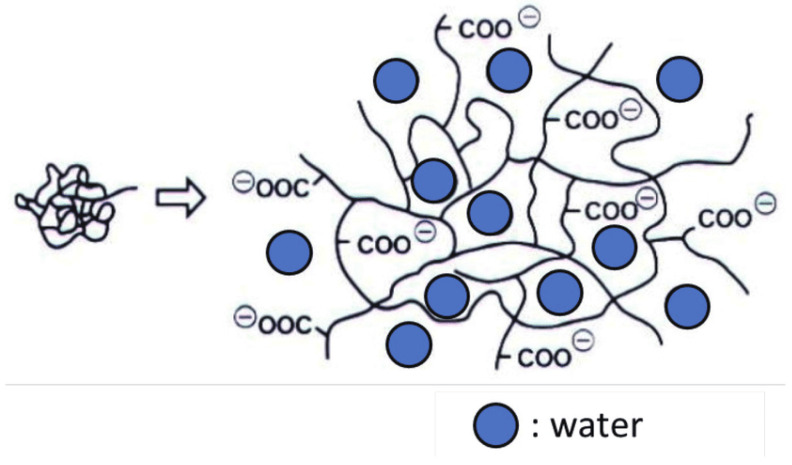
Water absorbing mechanism of superabsorbent polymer.

**Figure 3 polymers-15-00161-f003:**
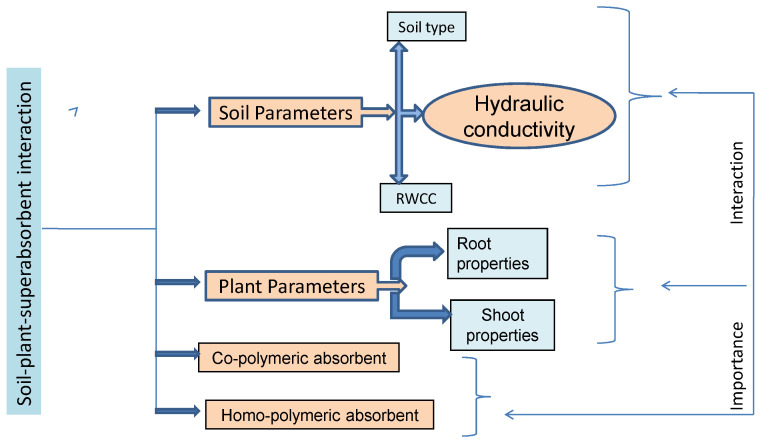
Important aspects of soil plant-super absorbent interaction are to be explored for drought management.

**Table 1 polymers-15-00161-t001:** In India water demand and availability (Report, NCIWRD,1999).

Year	Groundwater (BCM)	Water Demand (BCM)	Per Capita Water Availability (m^3^/Year)	Water Requires Irrigation (BCM)
2011	433	813	1545	557
2025		1093	1340	611
2050		1447	1140	817

Source: Basin Planning Directorate, CWC, XI Plan Document.

**Table 2 polymers-15-00161-t002:** Irrigation time interval under hydrogel and control *.

Irrigation Duration under Control (DAS)	Irrigation Duration under Hydrogel (DAS)
20–22	25
40–45	50
70–75	-
90–95	105
110–115	-

* Akhter, et al., 2004.

**Table 3 polymers-15-00161-t003:** Agricultural hydrogel products in India.

Trade Name	Manufacturing Groups
Pusa hydrogel	IARI, New Delhi, India
Waterlock 93 N	Acura Organics Ltd., New Delhi
Agroforestry water absorbent polymer	Technocare products, Ahmedabad,
Super absorbant polymer	Gel Frost packs Kalyani Enterprises, Chennai
Hydrogel	ChemtexSpeciality Ltd., Mumbai
Raindrops	M5 Exotic Lifestyle Concepts, Chennai

**Table 4 polymers-15-00161-t004:** Summary of relevant work related to SAP influenced crop growth and yield.

References	Crop	SAP Application	Conclusion
Saini et al. [1]	Pearl millet	2.5 kg/ha 5 kg/ha	2.32 and 4.37% earhead girth, 6.15 and 12.73% test weight increased over control respectively to both treatments
Choudhary et al. [107]	Mustard	2.5 kg/ha 5 kg/ha	The application of 5.0 kg/ha under moisture stress and 2.5 kg/ha under normal moisture was beneficial for the production
Mazloom et al. [3]	Maize	Lignin hydrogel	Lignin hydrogel increased water availability in maize and enhance the P uptake
Islam et al. [24]	Wheat		Root shoot growth, and yield increased while leaf area and chlorophyll index were not affected
Jnanesha et al. [108]	Seena	2.5 kg/ha 3.0 kg/ha	In order to increase soil water retention and absorption, combat water shortages, and lessen the negative consequences of drought stress, hydrogel is essential and reported significantly more leaves and pods
Ahmed et al. [109]	corn	Stockosorb 660, (1.0–2.5 mm), Hydrosource, (1.0–2.0 mm) SuperAB A200 (>2–5 mm)	The performance of WAC, re-swelling capacity, and significantly improved WHC of the soil was more reliant on water salinity and concentration than SAP type

**Table 5 polymers-15-00161-t005:** Soil properties are affected by the application of super-absorbent polymers.

Crop-Soil Type and Properties	Conclusion	Reference
Wheat- BD, Total porosity, WHC, pH, DA, and total Bacterial Count	It has the potential to improve the soil-like porosity (0.26 to 6.91%), water holding capacity (5.68–17.90%)	Ashari et al. [121]
Senna- WHC, soil moisture availability, porosity, and BD	Asp increased WHC and Soil moisture availability while there was no effect on porosity and BD.	Jnanesha et al. [108]
Sandy clay loam- soil moisture retention	In comparison to untreated soils, the insertion of Pusa Hydrogel improved the soil’s capacity to hold and keep more moisture, with consistent and gradual release of soil moisture necessitating less and less frequent watering.	Nutan Kujur et al. [122]
Pea nut-	Improve productivity and also water use efficiency	Jain et al. [123]
Maize	More than SAP type, water salinity, and concentration had a greater impact on WAC, re-swelling capacity, and the soil’s WHC.	AbdAllah et al. [109]

## Data Availability

Not applicable.

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
