# Peer review of "Superabsorbent Polymers as a Soil Amendment for Increasing Agriculture Production with Reducing Water Losses under Water Stress Condition"

_polymers, 2022, doi:10.3390/polym15010161_

Round 1

Reviewer 1 Report

This study reviewed the use of superabsorbent polymers (SAP) as amendment for soil health improvement ultimately improves water holding capacity and plant available water. I recommend the manuscript for publication after considering the following suggestions which their addressing will fit the manuscript for publication.

Comments

1.     Abstract needs to include some scientific and numerical data to catch the readers. Now it's just a description

2.     At the end of introduction state, the clear objectives of this manuscript.

3.     Introduction must state the need and novelty of this manuscript

4.     The conclusion is very short, please add some conclusive results.

5.     Important researches are missing in the manuscript. Please refer and cite the following references: Clean. Eng. Technol. 4 (2021) 100209; Nanotechnol. Environ. Eng. 6 (2021) 9.

6.     Please check the grammar, uniformity in the reference format, and spell-check are necessary throughout the manuscript.

Author Response

Pointwise answers to reviewer’s comments

English language and style are fine/minor spell check required: English language and style are needful done as highlighted and minor spell check is also needful done.

Comments and Suggestions for Authors

S.No

Comment

Reply

Reviewer #1:

1.

Abstract needs to include some scientific and numerical data to catch the readers. Now it's just a description.

The present manuscript is a review not a research article. So scientific and numerical data is not added.

Kindly refer line no. 31-47.

2.

At the end of introduction state, the clear objectives of this manuscript.

The objectives of the manuscript is added.

Kindly refer line no. 103-109.

3.

Introduction must state the need and novelty of this manuscript

Needful done.

4.

The conclusion is very short, please add some conclusive results.

Needful done.

Kindly refer line no. 561-569.

5.

Important researches are missing in the manuscript. Please refer and cite the following references: Clean. Eng. Technol. 4 (2021) 100209; Nanotechnol. Environ. Eng. 6 (2021) 9.

The following citations have been added in the manuscript.

Kindly refer line no. 67-68 and 253-254.

6.

 Please check the grammar, uniformity in the reference format, and spell-check are necessary throughout the manuscript.

The grammar, uniformity in the reference format, and spell-check are needful done.

Reviewer 2 Report

Figure 1 needs to be reformatted to show the contents. 

There are 2 Table2. Please correct the TABLE number. 

Section 4 is not relevant to the topic of this review. I suggest to remove it, which can help the audience focus on the main idea. 

The examples and referred work in this review is relatively old (<20% reference are within 5 years). I strongly suggest the authors to look into research work that is newer and give an updated review on this topic. 

Author Response

Pointwise answers to reviewer’s comments

English language and style are fine/minor spell check required: English language and style are needful done as highlighted and minor spell check is also needful done.

Comments and Suggestions for Authors

S.No.

Comment

Reply

Reviewer #2:

1.

Figure 1 needs to be reformatted to show the contents. 

Figure 3 (Figure 1 old number) is reformatted as per the suggestions.

Kindly refer line no. 537-538

2.

There are 2 Table2. Please correct the TABLE number. 

Corrected the Table numbers.

Kindly refer line no. 135 for Table 1 and 145 for Table 2.

3.

Section 4 is not relevant to the topic of this review. I suggest to remove it, which can help the audience focus on the main idea. 

Section 4 has been deleted as suggested.

Kindly refer line no. 256-277.

4.

The examples and referred work in this review is relatively old (<20% reference are within 5 years). I strongly suggest the authors to look into research work that is newer and give an updated review on this topic. 

Needful done. New references have been added throughout the manuscript.

Reviewer 3 Report

This review introduces the recent advances on the superabsorbent polymers used for soil amendment. The topic of this review is interesting.

Here are some suggestions for improving this review:

Abstract has introduced too many background information. However, the superabsorbent polymers are seldom introduced.

Keywords: Add some more keywords.

At the end of introduction, it is necessary to introduce the main focus of this review.

"In India water stress Status and Irrigation": It is suggested to introduce the status of the whole world not only restricted to India.

The superabsorbent polymers are not clearly introduced in this review. It is suggested to add a separate section to introduce superabsorbent polymers in details, such as the definition, preparation methods, properties, functionalities and applications. So section 3-7 should be rearranged.

The review does not contain interesting and attractive figures.

Author Response

Pointwise answers to reviewer’s comments

Extensive editing of English language and style required: English language and style are needful done as highlighted and minor spell check is also needful done.

Comments and Suggestions for Authors

S.No

Comment

Reply

Reviewer #3:

1.

Abstract has introduced too many background information. However, the superabsorbent polymers are seldom introduced.

The abstract has been rewritten as suggested.

Kindly refer line no. 28-47.

2.

Keywords: Add some more keywords.

Added more keywords.

Kindly refer line no. 49-50.

3.

At the end of introduction, it is necessary to introduce the main focus of this review.

The main focus of this review has been added at the end of introduction.

Kindly refer line no. 102-109.

4.

"In India water stress Status and Irrigation": It is suggested to introduce the status of the whole world not only restricted to India.

The status of the whole world has been added

Kindly refer line no. 111-123.

5.

The superabsorbent polymers are not clearly introduced in this review. It is suggested to add a separate section to introduce superabsorbent polymers in details, such as the definition, preparation methods, properties, functionalities and applications. So section 3-7 should be rearranged.

Rearranged the sections as suggested.

Kindly refer line no. 147-247.

6.

The review does not contain interesting and attractive figures.

New figures have been added in the manuscript.

Kindly refer line no. 57-160 and 201-2031.

Round 2

Reviewer 2 Report

The authors have addressed all the comments properly. 

Reviewer 3 Report

The revised manuscript is acceptable.